# Air-Pressure-Supported Application of Cultured Human Keratinocytes in a Fibrin Sealant Suspension as a Potential Clinical Tool for Large-Scale Wounds

**DOI:** 10.3390/jcm11175032

**Published:** 2022-08-27

**Authors:** Celena A. Sörgel, Rafael Schmid, Annika Kengelbach-Weigand, Theresa Promny, Raymund E. Horch

**Affiliations:** Department of Plastic and Hand Surgery and Laboratory for Tissue Engineering and Regenerative Medicine, University Hospital of Erlangen, Friedrich-Alexander University Erlangen-Nürnberg [FAU], 91054 Erlangen, Germany

**Keywords:** keratinocytes, fibrin, air pressure, wound healing, spray application

## Abstract

The treatment of large-scale skin wounds remains a therapeutic challenge. In most cases there is not enough autologous material available for full coverage. Cultured epithelial autografts are efficient in restoring the lost epidermal cover; however, they have some disadvantages, such as difficult application and protracted cell cultivation periods. Transplanting a sprayed keratinocyte suspension in fibrin sealant as biological carrier is an option to overcome those disadvantages. Here, we studied different seeding techniques regarding their applicability and advantages on cell survival, attachment, and outgrowth in vitro and thereby improve the cell transfer to the wound bed. Human primary keratinocytes were suspended in a fibrin sealant. WST-8 assay was used to evaluate the vitality for 7 days. Furthermore, the cells were labeled with CellTracker™ CM-Di-I and stained with a life/dead staining. Cell morphology, shape, and distribution were microscopically analyzed. There was a significant increase in vitality while cultivating the cells in fibrin. Sprayed cells were considerably more homogenously distributed. Sprayed cells reached the confluent state earlier than dripped cells. There was no difference in the vitality and morphology in both groups over the observation period. These findings indicate that the sprayed keratinocytes are superior to the application of the cells as droplets. The sprayed application may offer a promising therapeutic option in the treatment of large chronic wounds.

## 1. Introduction

The skin resembles the body’s interface with the external world and its outermost protection layer. It functions as a shield to external stress and pathogens and is essential for hydration and the thermal regulation of the human body. Physiological wound healing and skin regeneration is a complex process, which is essential for keeping the body’s homeostasis up after injuries. It involves an interplay between growth factors, cytokines, proliferating skin cells, and the body’s immune system. However, the skin is vulnerable to certain external damages, such as severe burns, chemical trauma, or skin diseases, such as chronic ulceration. In those conditions, the skin’s capacity to restore itself is reduced, which may lead to extensive deep or full-thickness skin defects [1]. In most cases, a therapeutic intervention is needed to restore skin continuity. The gold-standard treatment of large full-scale skin wounds is defined by the transplantation of autologous tissue. This includes skin grafts or tissue flaps. In extensive cases, such as severe or extensive third-degree burns, therapeutic alternatives are needed due to a lack of uninjured autologous tissue [2]. The amplification of human keratinocyte cells by serial cultivation can provide large quantities of epidermal cells, which have the potential to restore the vital functions of the epidermis and dermis. Commercially available CEAs have been obtainable for almost three decades and are known to be efficient in restoring lost epidermal cover. Nevertheless, there are several disadvantages, especially in terms of application and protracted cell cultivation periods. To overcome these issues, various alternative carrier substances that permit an easy application of autologous keratinocytes, such as natural or synthetic biomaterials, have been tested and developed [3,4].

The construction of composite grafts is one of the attempts to solve some of the well-known problems of this method. Several factors are responsible for a successful transplantation of human keratinocytes to a wound bed [5,6,7,8]. Among others, the type of the carrier material and transplantation technique, the number and age of the donor cells, the donor site, the age and overall condition of the patient, and the distribution of the cells on the wound bed or on the carrier are important issues [9,10,11].

Different biomaterials, such as gelatin, chitosan, or collagen, have been used to culture and transplant human keratinocytes [12,13,14]. Encouraging results were obtained in several studies using a suspension of human keratinocytes in fibrin sealant, which represents an important biopolymer in the skin’s recovery process from large wounds [2,15,16,17,18]. It has been proven to be a suitable cell carrier that shows high tissue tolerance properties because it is completely resorbed [19,20,21]. The role of fibrin as a universal component of the healing cascade in every wound has been widely described. It represents a physiological matrix, which plays an important role in hemostasis and induces the migration of endothelial cells [22,23,24]. In addition, it stimulates cell migration and thereby aids the arrival and adhesion of the cells at the level of the wound bed [25].

However, the ideal delivery and seeding of cultured cells to the recipient site is an unsolved problem. Cells, which are seeded as droplets onto a biomaterial or suspended in fibrin sealant, are not necessarily equally distributed over the maximally possible area, and may not be able to form a uniform layer on the surface. This may be a disadvantage for wound healing [26,27]. Questions have been raised as to whether a more equal distribution of cultured cells to larger wound surfaces could be achieved using air pressure spray systems without damaging the cultured keratinocytes by the mechanical impact of this special type of application. In addition to the improved distribution, the surgical handling could be simplified by spraying the keratinocytes, and the rate of cell yield could be potentially considerably increased when compared with droplet application. The aim of this study was to compare the distribution and morphology of human keratinocytes, which were suspended in fibrin sealant and seeded either conventionally or by using air pressure.

## 2. Materials and Methods

### 2.1. Patients

We used human skin from the abdomen for keratinocyte isolation. Tissue harvest was in conformity with the ethics commission of the Friedrich-Alexander University of Erlangen-Nürnberg (FAU) (Erlangen, Germany) (Ethics number 264_13B). The cell isolation was approved by the university and was carried out according to the World Medical Association Declaration of Helsinki’s policies.

### 2.2. Isolation and Cultivation of Human Keratinocytes

We used the Human Epidermis Dissociation Kit (Miltenyi Biotec B.V. & Co. KG, Bergisch-Gladbach, Germany) containing the Enzymes G, P, and A. We separated the skin from the subcutaneous fat tissue and cut it into small pieces of 16 mm^2^ in size, transferred into a mixture of RPMI 1640 Medium (Roswell Park Memorial Institute Medium 1640) (Thermo Fisher Scientific, Waltham, Massachusetts, USA) and Enzyme G, and incubated under constant rotating movements at 4 °C for 18 h. Afterwards, we separated the epidermis from the dermis and cultivated at 37 °C for 60 min in a mixture of Enzyme P and A. We placed the epidermis inside the gentleMACS™ C Tube on the gentleMACS™ Octo Dissociator (Miltenyi Biotec B.V. & Co. KG, Bergisch-Gladbach, Germany ). Then, we ran program B. In the next step, we filtered (70 µm pore size), centrifuged (300× *g*, 10 min), and resuspended the keratinocytes in Keratinocyte Growth Medium (KGM) (PromoCell GmbH, Heidelberg, Germany). We supplemented KGM with CaCl_2_ (0.06 mM), 10 µg/mL recombinant human transferrin-5, 0.33 µg/mL hydrocortisone-165, 0.39 µg/mL epinephrine-195, 0.004 mL/mL bovine pituitary extract, 0.125 ng/mL recombinant human epidermal growth factor, 5 µL/mL recombinant human insulin, 0.33 µg/mL hydrocortisone-165, 0.39 µg/mL epinephrine-195 (all from PromoCell GmbH, Heidelberg, Germany), and 1% penicillin–streptomycin (Sigma-Aldrich Corporation, St. Louis, MO, USA). We incubated the cells at 37 °C, 5% CO_2_ in cell culture flasks (T75) coated with 3.0 mg/cm^2^ rat tail collagen type I (Sigma-Aldrich Corporation, St. Louis, MO, USA). After an incubation period of 10–12 days, we split the cells at a confluence of 70–80%. We cultivated the cells up to passage 4.

### 2.3. Preparation of Keratinocyte–Fibrin–Sealant–Suspension

We diluted the human fibrinogen and thrombin components of a fibrin sealant kit (Tisseel^®^, Baxter International, Deerfield, IL, USA) 1:2 in KGM (final fibrinogen concentration 22.75 mg/mL, final thrombin concentration 125 I.U./mL). We resuspended keratinocytes in fibrinogen to achieve a final cell concentration of 4.0 × 105 cells/mL fibrin. We filled both components into separate chambers of a dual-chamber syringe under sterile conditions. During the ejection process, we interspersed and expulsed the two components as chelated hydrogel. We coated all wells with 3.0 mg/cm^2^ rat tail collagen type I (Sigma-Aldrich Corporation, St. Louis, MO, USA).

### 2.4. CellTracker™ CM-Di-I Staining

We cultured keratinocytes in KGM, as described above. After reaching a confluence of 60–70%, we detached the cells from the cell culture flask. We incubated 8 × 105 cells in 2.5 µg/mL CellTracker™ CM-Di-I (Molecular Probes Inc., Eugene, OR, USA) in KGM for 15 min at 37 °C. We washed cells twice with PBS and centrifuged (300× *g*, 4 min).

### 2.5. Seeding Procedure

We suspended the DiI-labeled keratinocytes in the fibrinogen component of the fibrin glue kit, as described above. We applied keratinocytes either as droplet (group I) or sprayed with pressurized air (group II). We seeded the cells in triplicates.

For preparation of the droplet group (I.), we applied a uniform droplet of 0.5 mL volume on the culture surface of a 6-well plate. We applied the spray group (II.) using a pneumatic atomizer and tube, creating compressed air. We connected the dual-chamber syringe to the compressed air system and sprayed 0.5 mL of fibrin from 3 cm distance using a pressure of 1.5 bar on a 6-well plate sprayed. We performed technical triplicates. After 15 min, we filled the wells with 3 mL KGM and incubated at 37 °C in a humidified atmosphere of 5% CO_2_.

### 2.6. Keratinocyte Vitality in Fibrin

We applied triplicates of 0.4 mL fibrin-cell-suspension droplets for each time point onto a 12-well plate. We evaluated the cell vitality using WST-8 assay (PromoCell GmbH, Heidelberg, Germany). We chose time points 1 and 7 days after seeding for assessment. We prepared WST-8 reagent solution by mixing CCVK1 reagent and cell culture medium in a ratio of 1:11. We added 500 µL of the solution to each well and incubated for 4 h at 37 °C and 5% CO_2_. We prepared a blank sample using a well without cells. We measured absorbance at 450 and 600 nm (background) with Multiscan Go and SkanIT RE for Multiscan Go 6.1 (Thermo Fisher Scientific, Waltham, MA, USA). We normalized vitality data to time point 0 h set as 1.

### 2.7. Microscopic Examination

We used Olympus IX83 (CellSens software, Olympus Corporation, Shinjuku, Tokyo, Japan) for microscopic examination. We examined samples on days 1 and 7 in 40- and 100-fold magnification using the tetramethylrhodamine (TRITC) and phase contrast channels.

On day 8, we stained the cells with calcein-AM/propidium iodide for microscopic analysis of cell survival. We washed the cells with Hanks Balanced Saline Solution (HBSS, Sigma-Aldrich Corporation) for 15 min at 37 °C and 5% CO_2_. We incubated keratinocytes in 2 µM calcein-AM (Sigma-Aldrich Corporation) for 30 min at 37 °C and 5% CO_2_. Subsequently, we stained the cells with 1.5 µM propidium iodide (Sigma-Aldrich Corporation) for 5 min at room temperature protected from light, washed in HBSS and immediately analyzed at 40- and 100-fold magnification using the tetramethylrhodamine (TRITC) and the fluorescein isothiocyanate (FITC) channels (Olympus IX83, cellSens software Shinjuku, Tokyo, Japan).

### 2.8. Area of Gel Distribution

On day 7, we took overview images of the wells in the TRITC channel. We quantified the area of gel distribution by measuring the cell-free area of the well with Fiji Is Just ImageJ (Fiji) 1.51 u (extended distribution). We expressed the values in cm^2^.

### 2.9. Statistical Analysis

We analyzed the results statistically with Graph Pad Prism 7 (GraphPad Software, Version 9.3.1, Inc., San Diego, CA, USA) using the *t*-test. We used mean ± standard deviation to present the data. *p* values ≤ 0.05 defined statistical significance. We adjusted images by PowerPoint (Microsoft Cooperation, Redmond, WA, USA) for better visibility of contrast and brightness.

## 3. Results

This section is divided by subheadings and provides a concise and precise description of the experimental results and our interpretation of them, as well as the experimental conclusions that can be drawn.

### 3.1. Cell Vitality

Keratinocyte vitality was measured at days 1 and 7 as triplicates to analyze the survival and replication of the cells inside the fibrin gel. The cells showed a significant increase in the number of living cells after a 7-day incubation period (Figure 1).

### 3.2. Morphology and Growth

The sprayed cells were uniformly distributed inside the hydrogel 1 d after inoculation, and the gel was uniformly distributed inside the cell culture well. Dripped keratinocytes were accumulated in and around the clots of fibrin sealant. The clots of fibrin were not equally distributed inside the well (Figure 2).

During the whole observation period, there was no difference in the cell morphology of the human keratinocytes, whether sprayed by using air pressure or seeded as droplets (Figure 2 and Figure 3). In the spray group, the inoculation of air bubbles into the sprayed fibrin hydrogel have been observed.

The same amount of fibrin gel containing the cells in the sprayed group covered a significantly larger surface inside the well plate than the dripped cells (Figure 4A). On day 8 of the experiment, the cells were stained with calcein-AM (living cells) and propidium iodide (dead cells). Microscopically, there was a tendency towards better cell survival in the sprayed group when compared with the group applied as droplet. Differences in distribution could be clearly observed (Figure 4A). The sprayed keratinocytes were distributed more equally than the dripped group (Figure 4B,C).

## 4. Discussion

Because the cultured epithelial sheet grafts are associated with several disadvantages in terms of handling and attachment to the wound bed, many efforts have been made aimed at optimizing the transfer of human keratinocytes to the recipient wound [28]. In addition to combined two-step operations with a pretreatment of the wounds by allografting, composite grafts constructed in vitro have been used experimentally and clinically [29]. They are a crucial therapeutic option for large-scale wounds that cannot be repaired with tissue flaps and instead rely on skin transplants [30,31].

Our group reported the successful application of a keratinocyte fibrin sealant suspension to burn wounds combined with the conventional type of sandwich technique grafting procedure, and encouraging results have been obtained. As demonstrated by Horch et al. the usage of a fibrin–glue matrix as a carrier material enhances the formation of the dermo–epidermal junction [6,15,16]. Auger et al. have highlighted a positive effect on the attachment properties of cultured epithelial sheets in a rat model [32]. Hunyadi et al. demonstrated that keratinocytes in a fibrin net showed excellent attachment properties, resulting in a rapid healing of full-thickness skin defects caused by leg ulcers or burns [33]. In our study we were able to confirm excellent keratinocyte survival in fibrin hydrogels. However, there are concerns regarding whether a cell suspension of cultured keratinocytes can survive the physical stress of an air-supplied spray application to a wound bed [34]. Spray application would provide an effective tool for evenly distributing cultured cells to a given surface. Furthermore, it would simplify the surgical handling of such cell transplants. Aerosols of cultured cells have been successfully used by Navarro et al.; however, the keratinocytes were sprayed in culture medium [35]. There was successful utilization of sprayed keratinocytes and fibroblasts for treatment of chronic wounds. These studies demonstrated positive effects on wound healing and the aesthetic results. Nevertheless, there was no examination of the application method that compared the spraying of cells with dripping [36]. We aimed to create a suspension supernatant that would not only protect the cells during the spraying process but also support their vitality. The viscosity of fibrin sealant compared with culture medium or other suspensions seems to be an ideal carrier to protect keratinocytes during the spraying process and to alleviate the impact of the mechanical force during the delivery [34]. In our study we used an aerosol of fibrin sealant as a cell delivery system for highly proliferative human keratinocytes instead of producing an aerosol of human keratinocytes and culture medium alone. We were able to show in vitro that there was excellent human keratinocyte cell vitality in the fibrin sealant, and the cells were obviously not damaged when being seeded using an air pressure of 1.5 bar. In contrast to the conventionally seeded human keratinocytes, those cells formed a uniform layer on the cell culture surface from the inoculation onwards. Microscopically, the morphology of the cells in both groups did not differ during the whole observation period. The bottom of the culture systems reached the confluent state, whereas in the control groups, the cells did not visibly migrate. The dropped cells were more difficult to evaluate due to their partial embedding in fibrin glue, as shown in scanning electron micrographs.

It is obvious that cells that cover a wound uniformly will contribute to a quicker spread over the whole area and might well improve wound healing [37]. We previously showed the feasibility of using collagen membranes and hyaluronic acid to serve as appropriate substrates to grow and multiply vital human keratinocytes, which could provide possible carrier systems for human keratinocytes [5]. The use of monolayers of cultured keratinocyte transplanted upside-down to wounds is especially facilitated by utilizing such carrier materials. The combination application of keratinocytes in fibrin sealant can influence the wound healing process positively by reducing the migration of macrophages into the wound [6,17,38]. Spraying cells has been reported earlier but was hampered by a lack of immediate attachment to the recipient wound bed when liquid solutions were used. Fibrin sealant on the other hand is a suitable cell carrier [33]; however, it has not been thoroughly tested as a spray delivery system for cultured human keratinocytes.

This application method facilitates handling and guarantees the equal and optimized distribution und utilization of cultured keratinocytes. Particularly, in the treatment of large wounds, it offers a uniform and efficient application method. It might lead to the delivery of more evenly distributed cells to a wound and, hence, present a useful approach in the clinical therapy of burns and chronic wounds.

## 5. Conclusions

The sprayed application of human keratinocytes leads to better and more uniform distribution and a faster confluence of cultured human keratinocytes in vitro. Keratinocytes were not damaged when seeded using an air pressure of 1.5 bar. From the inoculation onward, the cells formed a uniform layer on the cell culture surface. The viscosity of fibrin sealant compared with culture medium seems to be an ideal carrier to protect keratinocytes during the spraying process. These findings indicate that the sprayed keratinocytes are superior to the application of the cells as droplets. The sprayed application may offer a promising therapeutic option in the treatment of large chronic wounds.

## Figures and Tables

**Figure 1 jcm-11-05032-f001:**
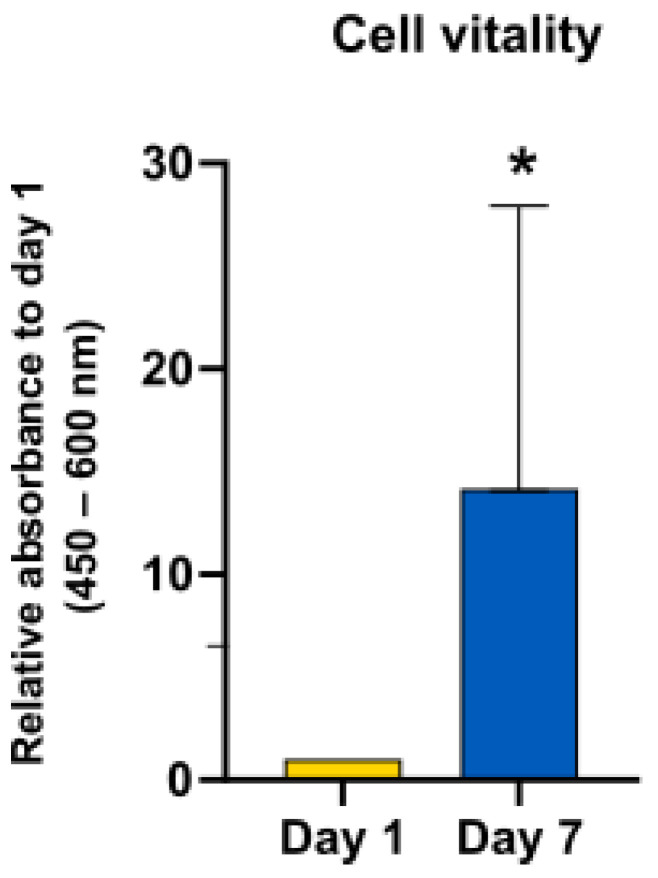
Keratinocyte vitality in fibrin on days 1 and 7. Sample size *n* = 3. Y-axis shows absorbance on 1 d and 7 d. Day 1 was normalized to 1. Average and standard deviation constituted. * *p* ≤ 0.05, significances were correlated to the relative absorbance on day 1.

**Figure 2 jcm-11-05032-f002:**
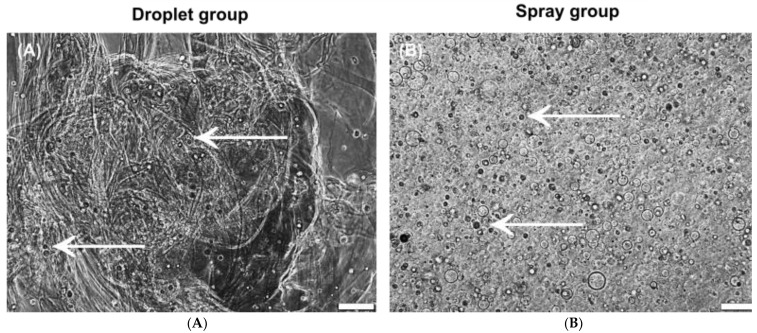
Gel structure and distribution of keratinocytes after droplet and spray application. (**A**,**B**) Representative images of dripped (**A**) and sprayed (**B**) keratinocytes at time point 1d, phase contrast. Scale bar 200 µm. ← for better cell demonstration and identification of cell morphology.

**Figure 3 jcm-11-05032-f003:**
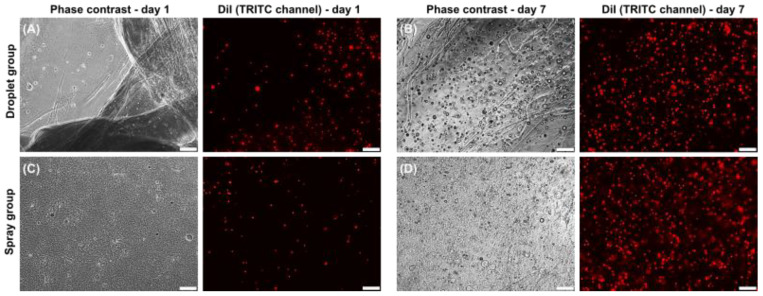
Keratinocytes after droplet and spray application. (**A**–**D**) Representative images of dripped (**A**) and sprayed (**B**) keratinocytes at time points 1 and 7 d, phase contrast and CellTracker™-CM-Di-I-labelled keratinocytes (red). Scale bar 200 µm.

**Figure 4 jcm-11-05032-f004:**
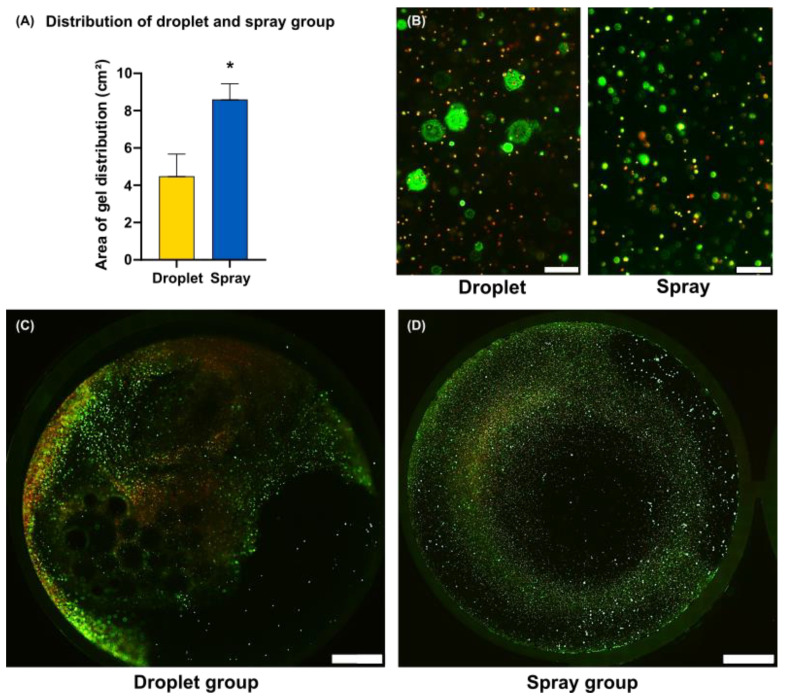
Distribution of keratinocytes after spray and droplet application. Sample size *n* = 3. (**A**) Y-axis shows surface in cm^2^ on 7 d. (**B**–**D**) Representative images of sprayed and dripped keratinocytes at time point 8 d, live/dead staining (green = living cells, red = dead cells). Scale bar (**B**) = 200 µm, (**C**,**D**) 5 mm. * *p* ≤ 0.05, significances were correlated to the relative surface area on day 1.

## Data Availability

Data available on request due to privacy and ethical restrictions. The data presented in this study are available on request from the corresponding author.

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
