# Peer review of "Air-Pressure-Supported Application of Cultured Human Keratinocytes in a Fibrin Sealant Suspension as a Potential Clinical Tool for Large-Scale Wounds"

_jcm, 2022, doi:10.3390/jcm11175032_

Round 1
Reviewer 1 Report
In this manuscript, the effect of seeding techniques on the cell fate of fibrin-suspended keratinocytes was evaluated. It is a relatively simple study comparing spayed cells with droplet seeding.
1. Cell viability: as I understand, the authors evaluated "viability" using WST-8 assay, which in fact, is a formazan-based technique. The amount of formazan produced is directly proportional to the number of living cells. Therefore, if measured on day 1 and day 7 (normalized to day 1) the result is a simple increase in cell number and not a measurement of viability. The number of cells is expected to increase over time. Based on this result, one can't conclude that the "cells showed a significant increase in viability after a 7-day incubation period." For viability study over time AnnV PI/7aad assay would be better. WST-8 assay could be used to compare sprayed vs droplet group at the same time point (if the number of the seeded cells was the same).
The idea of spraying is seems promising but the manuscript lacks proof and statistical significance of results.
Author Response
We thank the reviewer for the time and effort invested into the assessment of our research. We agree with the reviewer that the number of living cells does not equal cell viability. Therefore, throughout the manuscript the term “viability” was replaced by “vitality” and the significant increase in absorbance was clearly referred to the increase in cell number.
We accept the reviewer’s advice to use the AnnV PI/7aad assay thankfully and will include this in our further studies as a primary investigation.
Reviewer 2 Report
Sörgel et al., evaluated the feasibility of applying human keratinocytes in a fibrin sealant supension with the help of air pressure. The work is very interesting but it needs some improvements:
-Abstract: must be rewritten, be more specific and follow an order.
-Introduction: it is very brief. Authors should provide more informacion regarding the different methodologies in keratinocyte delivery in burn wound care. Indeed, they shoul include this ref: Bioact Mater. 2022 Oct; 16: 187–203
-Material and methods: It does not follow a meaningful order. For example, why the "seeding procedure" section is described after the "Keratinocyte viability in fibrin" section?
-Figure 1: it must be improved. What are the n samples? Th error bar does not make any sense. The asterisk represents P values ≤ 0.05?
Figure 2: what are the circles in the spray group figure? Bubbles??? Please , use arrows or other tools to indentify the cell morphology in both figures
Figure 3: Please improve the quality of the figures at day 1
Figure 4: Does the author prove that keratinocytes are attached? Figure 4 B does not seem that they are attached. Fibrin viscosity must be taken into account.
Authors should perfom a colony-forming assayor related study to demostrate that human primary keratinocyte stemness is not affected during the process.
Authors should justify why 1.5 bar is used.
Author Response
Thank you very much for the time and effort invested in reviewing our manuscript as an expert in the field. We hope to have improved our submission according to your suggestions.
-Abstract: must be rewritten, be more specific and follow an order.
We thank the reviewer for the time and effort invested into the assessment of our research. We agree with the reviewer and have rewritten the abstract to be more specific and structured (lines 10-32).
-Introduction: it is very brief. Authors should provide more informacion regarding the different methodologies in keratinocyte delivery in burn wound care. Indeed, they shoul include this ref: Bioact Mater. 2022 Oct; 16: 187–203
According to the reviewers suggestions we have added additional information regarding the different methodologies in cell delivery in the treatment of large scale skin injuries as for example in burn wound care. We have improved our reference list according to the reviewers wishes (lines 37 to 76).
-Material and methods: It does not follow a meaningful order. For example, why the "seeding procedure" section is described after the "Keratinocyte viability in fibrin" section?
We agree with the reviewer and have changed the order of the materials and methods subsection according to the reviewers wishes.
-Figure 1: it must be improved. What are the n samples? Th error bar does not make any sense. The asterisk represents P values ≤ 0.05?
We thank the reviewer for pointing out the improvements needed in Figure 1. We have added the specific sample size in line 158 and also specified the sample size in the legend of the figures (lines 202 and 247). We thank the reviewer for pointing out the mistake made by us regarding the error bar. We have corrected this throughout the manuscript.
Figure 2: what are the circles in the spray group figure? Bubbles??? Please , use arrows or other tools to indentify the cell morphology in both figures
Indeed the circles inside the images of figure 2 mentioned by the reviewer are bubbles. We have added this in the result section (line 226-227).During the spraying process the inclusion of a small amount of air inside the hydrogel was unavoidable. We have added arrows to clearly point out the cells in the images of figure 2.
Figure 3: Please improve the quality of the figures at day 1
We have made the according chances to improve the images. Also, we increased the image size to provide better resolution. We hope to have increased the image quality thereby.
Figure 4: Does the author prove that keratinocytes are attached? Figure 4 B does not seem that they are attached. Fibrin viscosity must be taken into account.
We understand the question arising from figure 4B that the cells might not seem attached.
Nevertheless, it has to be taken into consideration that a clear and sharp cell representation in a 3D model is a challenge. In our opinion it is the superposition inside the 3D hydrogel that creates the semblance the cells not to be attached. This superposition cannot be avoided by the imaging technique used in our studies. This leads to artefacts on the one hand and on the other hand makes a clear cell distinction impossible. This creates a major bias we were trying to avoid. For clear distinction, better imaging techniques, as for example confocal microscopy and 3D reconstruction of the images would be needed. We plan to include those in our future studies. Furthermore, the attachment of the cells in the fibrin hydrogels was demonstrated in our past manuscripts. We have improved our reference list accordingly and have addressed the problem in the discussion subsection (lines 268 to 279)
Authors should perfom a colony-forming assayor related study to demostrate that human primary keratinocyte stemness is not affected during the process.
We thank the author for the suggestion to improve our work by including a colony forming assay. Our aim regarding this manuscript was a comparison of the application methods regarding the delivery of the cells to the wound bed and their viability. The evaluation of changes in the keratinocyte stemness after the spraying process is a very interesting matter, which unfortunately was out of the scope of this study to evaluate. We hope to be addressing the matter during the further course of our research in this field.
Authors should justify why 1.5 bar is used.
1.5 bar was the lowest of the preset settings of the airpressure device available in our laboratory. Since with this pressure we achieved a good distribution of the sprayed sealant and we aimed to avoid cell damage caused by higher pressure levels, the experiment was carried out with 1.5 bar.
Reviewer 3 Report
Very interesting study. But I could not find the sample size of group I and II. This is of outmost importance and is missing? Please add.
Author Response
We thank the reviewer for pointing out the lacking sample sizes. We have added the specific sample size in line 158 and also specified the sample size in the legend of the figures (lines 202 and 247). We have removed references with a lack of relevance to the research.
Round 2
Reviewer 1 Report
The authors introduced revisions in the article, however, I still believe that the 3.1 cell vitality paragraph doesn't provide any information. Of course, there will be more cells after 7 days (unless they die due to any reason). This would only be meaningful if the number of cells (using the WST assay) is compared in the spray and droplet groups.
The number of seeded cells must be the same. Test performed after one day will show the influence of the seeding technique on the number of cells (do cells die when sprayed?), and a test performed after 7 days will show if the spray technique and resulting more uniform distribution of cells leads to faster proliferation and thus higher number of cells then standard droplet method.
Author Response
We thank the reviewer for the time and effort invested into the assessment of our research. Since primary human keratinocytes, which were used for this study, are vulnerable to external influences, our aim in paragraph 3.1 regarding the cell vitality was to evaluate whether the fibrin spray used for our experiments would be a suitable scaffold for the cells.
Indeed, we agree with the reviewer that a comparison of cell vitality of the spray and droplet technique would be an interesting addition to our research. This experiment was conducted with a WST-8 assay, yet the color change was not quantifiable in both groups on day 1, as it was too low. We believe that because of the larger surface area of the hydrogel inside the 6 well-plate the diffusion of the reagent into the hydrogel was compromised. Spraying the cells on a smaller culture plate would have been improper. The smallest possible spray radius while using 1.5 bar was around 10cm² which resembles the surface area of the plate used for our experiments. Using smaller plates would have compromised seeding the same number of cells due to the cell loss around the culture plate. Therefore, we have used a live-dead staining technique and have microscopically displayed the entire well to show there is no significant microscopic difference between the spray and droplet group regarding living and dead cells.
Nevertheless, we in our future work we will include a vitality study over time with a comparison of both groups. According to the reviewer’s suggestion from the first review round we will use other examination techniques such as AnnV PI/7aad assay and hope to be more successful.